# Hospital-Acquired Serum Chloride Derangements and Associated In-Hospital Mortality

**DOI:** 10.3390/medicines7070038

**Published:** 2020-06-29

**Authors:** Charat Thongprayoon, Wisit Cheungpasitporn, Tananchai Petnak, Michael A. Mao, Api Chewcharat, Fawad Qureshi, Juan Medaura, Tarun Bathini, Saraschandra Vallabhajosyula, Kianoush B. Kashani

**Affiliations:** 1Division of Nephrology and Hypertension, Department of Medicine, Mayo Clinic, Rochester, MN 55905, USA; api.che@hotmail.com (A.C.); qureshi.fawad@mayo.edu (F.Q.); kashani.kianoush@mayo.edu (K.B.K.); 2Division of Nephrology, Department of Internal Medicine, University of Mississippi Medical Center, Jackson, MS 39216, USA; jmedaura@umc.edu; 3Division of Pulmonary and Critical Care Medicine, Faculty of Medicine, Ramathibodi Hospital, Mahidol University, Bangkok 10400, Thailand; petnak@yahoo.com; 4Division of Pulmonary and Critical Care Medicine, Department of Medicine, Mayo Clinic, Rochester, MN 55905, USA; 5Division of Nephrology and Hypertension, Mayo Clinic, Jacksonville, FL 32224, USA; mao.michael@mayo.edu; 6Department of Internal Medicine, University of Arizona, Tucson, AZ 85719, USA; tarunjacobb@gmail.com; 7Department of Cardiovascular Medicine, Mayo Clinic, Rochester, MN 55905, USA; Vallabhajosyula.Saraschandra@mayo.edu

**Keywords:** hyperchloremia, hypochloremia, chloride, electrolytes, internal medicine, mortality, nephrology, hospitalization

## Abstract

**Background:** We aimed to describe the incidence of hospital-acquired dyschloremia and its association with in-hospital mortality in general hospitalized patients. **Methods:** All hospitalized patients from 2009 to 2013 who had normal admission serum chloride and at least two serum chloride measurements in the hospital were studied. The normal range of serum chloride was defined as 100–108 mmol/L. Hospital serum chloride levels were grouped based on the occurrence of hospital-acquired hypochloremia and hyperchloremia. The association of hospital-acquired hypochloremia and hyperchloremia with in-hospital mortality was analyzed using logistic regression. **Results:** Among the total of 39,298 hospitalized patients, 59% had persistently normal hospital serum chloride levels, 21% had hospital-acquired hypochloremia only, 15% had hospital-acquired hyperchloremia only, and 5% had both hypochloremia and hyperchloremia. Compared with patients with persistently normal hospital serum chloride levels, hospital-acquired hyperchloremia only (odds ratio or OR 2.84; *p* < 0.001) and both hospital-acquired hypochloremia and hyperchloremia (OR 1.72; *p* = 0.004) were associated with increased in-hospital mortality, whereas hospital-acquired hypochloremia only was not (OR 0.91; *p* = 0.54). **Conclusions:** Approximately 40% of hospitalized patients developed serum chloride derangements. Hospital-acquired hyperchloremia, but not hypochloremia, was associated with increased in-hospital mortality.

## 1. Introduction

Chloride ions are the most dominant anions in the extracellular fluid. The intracellular chloride concentration varies, and it can range from 2 mmol/L in skeletal muscle to 90 mmol/L in erythrocytes [1]. Chloride ions serve principal roles in maintaining several physiologic functions, including the acid-base balance, hydrochloric acid production in the stomach, and cellular electrolyte regulation [2]. Although chloride is one of the major electrolytes reported in the basic chemistry panel, physicians often overlook the importance of dyschloremia in clinical practice. Hospital-acquired dyschloremia is among the common electrolyte abnormalities, ranging from 30% to 40% of hospitalizations [3,4,5,6,7,8]. The impact of hyperchloremia on renal function has been particularly considered for several years. Previous have studies revealed that hyperchloremia resulted in renal vasoconstriction through tubuloglomerular feedback and an increase of thromboxane release [9,10]. Recent data has demonstrated that dyschloremia is associated with the worse patient outcomes, including acute kidney injury and longer hospital lengths of stay [3,4,5,6,11,12,13,14]. 

While hypochloremia, in most cases, is community-acquired, hyperchloremia is often a hospital-acquired entity [4]. Patients who present with normochloremia at admission may later develop dyschloremia during their hospitalization. Hospital-acquired alterations of chloride homeostasis may result as either a complication from diseases or treatments prescribed by physicians. The administration of chloride-rich crystalloids is one of the most common causes of iatrogenic hospital-acquired hyperchloremia, particularly in critically ill patients. Several animal and human studies have demonstrated that an infusion of 0.9% saline is associated with increased serum chloride levels [6,15,16,17]. Other hospital-acquired etiologies for dyschloremia include hyperchloremia induced by hospital diarrhea or diabetes insipidus and hypochloremia associated with diuretic use, congestive heart failure, nasogastric tube drainage, and vomiting [1]. 

The currently available literature regarding dyschloremia impacts on patient outcomes has been primarily focused on critically ill patients and admission serum chloride levels. Data on hospital-acquired dyschloremia has been limited. Therefore, this study aimed to describe the incidences of hospital-acquired dyschloremia and their associations with in-hospital mortality in hospitalized patients in general wards.

## 2. Materials and Methods

### 2.1. Study Population

This study was approved by the Mayo Clinic Institutional Review Board (IRB number: 15-000024; Approval Date: 4 February 2015) and exempted from informed consent due to its minimal-risk nature. The institutional patient database was searched to identify adult patients admitted to Mayo Clinic Hospital, Rochester, MN, USA from 1 January 2011 to 31 December 2013. The objective of this study was to examine the incidences and impacts of hospital-acquired serum chloride derangements. Only patients with a normal serum chloride level at the time of hospital admission and at least two measured hospital serum chloride values were included in this study. For patients with recurring admissions, only the first hospitalization during the study period was included for analysis.

### 2.2. Definition of Hospital-Acquired Dyschloremia 

The normal range of serum chloride was defined as 100–108 mmol/L, as previously described [6]. Hospital-acquired hypochloremia was identified when the lowest serum chloride value during hospitalization was below 100 mmol/L. In contrast, hospital-acquired hyperchloremia was identified when the highest serum chloride value during hospitalization was above 108 mmol/L. Patients were categorized into four groups based on the occurrence of hospital-acquired dyschloremia: (1) persistently normal serum chloride levels, (2) hospital-acquired hypochloremia only, (3) hospital-acquired hyperchloremia only, and (4) both hospital-acquired hypochloremia and hyperchloremia. 

### 2.3. Covariates

Clinical information and laboratory data were extracted from Mayo Clinic’s institutional electronic medical record system. All serum chloride values measured during hospitalization were reviewed. Principal diagnoses were grouped based on the International Classification of Diseases, 9th Revision (ICD-9) codes. The comorbid conditions were abstracted using a previously validated data algorithm [18], and the comorbidity burden was summarized using the Charlson comorbidity index [19]. The baseline glomerular filtration rate (eGFR) was estimated based on age, sex, race, and serum creatinine, using the Chronic Kidney Disease Epidemiology Collaboration (CKD-EPI) equation [20]. An acute kidney injury (AKI) was defined according to the KDIGO criteria as an increase in serum creatinine of ≥0.3 mg/dL or ≥1.5 times from the baseline value [21], while the admission serum creatinine was used as the baseline serum creatinine.

### 2.4. Outcomes

In-hospital mortality was the outcome of interest. Death status at the time of hospital discharge was obtained from the institutional database. 

### 2.5. Statistical Analysis

Continuous variables were summarized as the mean ± standard deviation (SD) or median with interquartile range (IQR) as appropriate, depending on the normality of their distribution. The differences in the continuous variables between the in-hospital serum chloride groups were tested using an analysis of variance (ANOVA). Categorical variables were summarized as the frequency with percentages. The differences in the categorical variables between the in-hospital chloride groups were tested using the chi-square test. A logistic regression analysis was performed to assess the association of the hospital-acquired dyschloremia with the in-hospital mortality, compared with persistently normal serum chloride levels. The analysis was adjusted for prespecified variables, including age, sex, race, principal diagnosis, comorbidities, eGFR, intensive care unit admission, number of hospital serum chloride measurements, length of hospital stay, and admission serum chloride. An acute kidney injury was not included in the list of adjusting variables, because it was hypothesized to be a mediator for higher dyschloremia-associated mortality. A two-tailed *p* value < 0.05 was regarded as statistically significant. All analyses were performed using JMP statistical software (Version 14.1.0; SAS Institute Inc., Cary, NC, USA).

## 3. Results

### 3.1. Incidence of Hospital-Acquired Hypochloremia and Hyperchloremia

During the study period, 76,719 hospitalized patients with available serum chloride measurement at hospital admission were screened. After excluding patients with only one serum chloride measurement during hospitalization (n = 18,839) and patients with abnormal admission serum chloride levels (n = 18,582), a total of 39,298 patients with normal serum chloride levels at hospital admission were studied. Fifty-four percent of enrolled patients were male. The mean age was 63 ± 17 years. Twenty-seven percent of patients needed ICU admissions during hospitalization. The median number of hospital chloride measurements was four (two to six), and the length of hospital stay was five (three to seven) days. The incidences of hospital-acquired hypochloremia and hyperchloremia were 26% and 20%, respectively. Fifty-nine percent had persistently normal serum chloride levels in the hospital, 21% had hospital-acquired hypochloremia only, 15% had hospital-acquired hyperchloremia only, and 5% had both hypochloremia and hyperchloremia. Patient clinical characteristics categorized by their in-hospital serum chloride levels are summarized in Table 1. 

### 3.2. Association of Hospital-Acquired Dyschloremias with In-hospital Mortality

Patients with hospital-acquired hypochloremia had higher in-hospital mortality compared to those without hospital-acquired hypochloremia (1.6% vs. 1.1%; *p* < 0.001) (Table 2). After an adjustment for confounders, however, patients with hospital-acquired hypochloremia were not significantly associated with in-hospital mortality, compared with those without hospital-acquired hypochloremia with an adjusted odds ratio of 0.88 (95% confidence interval or CI 0.71–1.10; *p* = 0.27).

Patients with hospital-acquired hyperchloremia had higher in-hospital mortality compared to those without hospital-acquired hyperchloremia (3.8% vs. 0.6%; *p* < 0.001) (Table 2). After an adjustment for confounders, patients with hospital-acquired hyperchloremia remained significantly associated with an increased in-hospital mortality, compared with those without hospital-acquired hyperchloremia, with an adjusted odds ratio of 2.50 (95% CI 2.01–3.12; *p* < 0.001).

The in-hospital mortality was 0.9% in patients with hospital-acquired hypochloremia only, 3.5% in patients with hospital-acquired hyperchloremia only, and 4.8% in patients with both hypochloremia and hyperchloremia, compared with 0.5% in patients with persistently normal serum chloride levels (*p* < 0.001) (Table 2). After an adjustment for confounders, patients with hospital-acquired hyperchloremia only and both hospital-acquired hypochloremia and hyperchloremia were significantly associated with increased in-hospital mortality, with adjusted odds ratios of 2.84 (95% CI 2.20–3.68; *p* < 0.001) and 1.72 (95% CI 1.20–2.47; *p* = 0.004), respectively. In contrast, hospital-acquired hypochloremia only was not associated with significantly increased in-hospital mortality, with an adjusted odds ratio of 0.91 (95% CI 0.67–1.23; *p* = 0.54).

## 4. Discussion

Our study is composed of a large adult general hospitalized patient cohort focusing on the impact of hospital-acquired dyschloremia on in-hospital mortality. This study demonstrated that the incidence of hospital-acquired dyschloremia is 41%. The development of hospital-acquired hyperchloremia, either with or without hospital-acquired hypochloremia, was associated with an increased in-hospital mortality risk. In contrast, the in-hospital mortality was not significantly higher in patients with hospital-acquired hypochloremia only, compared to persistently normochloremic patients. 

The impact of serum chloride derangements on patient outcomes has been studied for years. Although chloride is the major anion in extracellular fluid, current knowledge of its clinical impact has been limited to settings with critically ill patients or admission chloride levels. Admission hyperchloremia has been associated with an increased risk of adverse patient outcomes, including a higher incidence of acute kidney injury and longer hospital length of stay [5,6,11,12,14,22,23]. The effect of serum chloride level alterations during hospitalization on patient outcomes has been reported in prior studies. An upward change in serum chloride levels during hospitalization has been associated with an increased risk of in-hospital mortality and acute kidney injury, regardless of admission serum chloride levels [5,13]. A subgroup analysis of a study of critically ill patients with normochloremia at admission demonstrated similar results, where an upward change in serum chloride levels during the first 72 h after admission was associated with a higher 30-day mortality risk [3]. In contrast, a downward change in serum chloride levels did not demonstrate an increased risk of mortality [3,5,13]. Our study adds to the current literature by showing that hospital-acquired hyperchloremia is also associated with an increased risk of in-hospital mortality. This may be especially relevant in the setting of iatrogenic hyperchloremia due to the administration of chloride-rich intravenous fluids.

The proposed mechanisms for the adverse clinical effects of hyperchloremia include a hyperchloremia-induced glomerular filtration rate decrease and the generation of metabolic acidosis. A prior animal study demonstrated that chloride-rich fluid infusions are detected in the renal tubules by the macula densa, resulting in afferent renal vasoconstriction and a reduction of the GFR rate via the tubuloglomerular feedback mechanism [10]. In addition to the tubuloglomerular feedback mechanism, hyperchloremia can indirectly cause renal vasoconstriction by increasing the thromboxane release and renal vascular responsiveness to vasoconstrictors (e.g., angiotensin II) [9,24]. These pathophysiological mechanisms may play a role in the observed increased risk of acute kidney injury with hyperchloremia [5,14,23,25]. Hyperchloremic metabolic acidosis may be another possible pathophysiologic mechanism that leads to worse patient outcomes. According to the Steward approach, a strong ion difference (SID) is one of the three variables utilized to determine the maintenance of hydrogen ions in the acid-base balance [1]. Chloride, the dominant strong anion in plasma, becomes a principal factor in determining the SID and acid-base balance. An increase of serum chloride levels out of proportion to the serum sodium levels results in a decrease of the SID and an increase of hydrogen ions and metabolic acidosis [26]. The severity of hyperchloremic metabolic acidosis can be enhanced when accompanied by anion gap metabolic acidoses, such as from lactic acidosis or severe acute kidney injury, leading to a higher risk of mortality [6]. In addition to hyperchloremia effects on the GFR and acid-base balance, a study using an animal sepsis model demonstrated that hyperchloremia is associated with an increase in the inflammatory cytokine release [27]. Thus, the hyperchloremia-associated augmentation of the inflammatory pathway may worsen the severity of diseases associated with an increased risk of in-hospital mortality. 

There are some limitations to our study. Although our research is composed of a relatively large cohort, this is a retrospective study. Therefore, a causal relationship between hospital-acquired dyschloremia and in-hospital mortality cannot be determined. We were also unable to adjust for all potential confounding factors that might affect patient outcomes, such as serum sodium levels, the fluid balance state, acid-base balance, quantity and type of intravenous fluid administration, and medications. Finally, our study consisted of predominantly white patients. Thus, the generalizability of the results may be limited.

## 5. Conclusions

In conclusion, hospital-acquired dyschloremia is common, with an incidence of 41%. Hospital-acquired hyperchloremia, with or without hypochloremia, was associated with an increased in-hospital mortality.

## Figures and Tables

**Table 1 medicines-07-00038-t001:** Patient clinical characteristics.

Variables	All	Serum Chloride during Hospitalization				
		Normal	Only Hypochloremia	Only Hyperchloremia	Both Hypo- and Hyperchloremia	*p*-Value *
N	39,298	23,056	8402	5932	1908	
Age (year)	63 ± 17	62 ± 17	63 ± 17	64 ± 18	63 ± 16	<0.001
Male sex	21,368 (54)	12,424 (54)	4960 (59)	2861 (48)	1123 (59)	<0.001
Caucasian	36,671 (93)	21,552 (93)	7862 (94)	5491 (93)	1766 (93)	0.03
Principal diagnosis						<0.001
Cardiovascular	9785 (25)	5032 (22)	2524 (30)	1311 (22)	918 (48)	
Hematology/oncology	5750 (15)	3254 (14)	1518 (18)	759 (13)	219 (11)	
Infectious disease	1247 (3)	516 (2)	186 (2)	449 (8)	96 (5)	
Endocrine/metabolic	836 (2)	477 (2)	148 (2)	194 (3)	17 (1)	
Respiratory	1513 (4)	862 (4)	368 (4)	228 (4)	55 (3)	
Gastrointestinal	3907 (10)	2132 (9)	705 (8)	903 (15)	167 (9)	
Genitourinary	1165 (3)	613 (3)	171 (2)	345 (6)	36 (2)	
Injury and poisoning	6301 (16)	3754 (16)	1321 (16)	947 (16)	279 (15)	
Other	8794 (22)	6416 (28)	1461 (17)	796 (13)	121 (6)	
Charlson comorbidity score	1.9 ± 2.4	1.7 ± 2.3	2.1 ± 2.5	2.1 ± 2.4	1.7 ± 2.2	<0.001
Comorbidity						
Coronary artery disease	3322 (8)	1842 (8)	786 (9)	515 (9)	179 (9)	<0.001
Congestive heart failure	2808 (7)	1287 (6)	898 (11)	432 (7)	191 (10)	<0.001
Peripheral artery disease	1330 (3)	640 (3)	343 (4)	241 (4)	106 (6)	<0.001
Stroke	3127 (8)	1720 (7)	685 (8)	581 (10)	141 (7)	<0.001
Diabetes mellitus	8004 (20)	4284 (19)	2081 (25)	1226 (21)	413 (22)	<0.001
Chronic obstructive pulmonary disease	3393 (9)	1708 (7)	954 (11)	529 (9)	202 (11)	<0.001
Cirrhosis	1002 (3)	493 (2)	220 (3)	228 (4)	61 (3)	<0.001
eGFR (mL/min/1.73 m^2^)	77 ± 28	79 ± 26	76 ± 28	68 ± 31	71 ± 29	<0.001
Acute kidney injury	5333 (14)	1710 (7)	1810 (22)	1070 (18)	743 (39)	<0.001
Intensive care unit admission	10,736 (27)	3634 (16)	3086 (37)	2427 (41)	1589 (83)	<0.001
Number of serum chloride measurements	4 (2-6)	3 (2-4)	5 (3-8)	6 (4-9)	13 (8-24)	<0.001
Length of hospital stay (day)	5 (3-7)	4 (3-6)	6 (4-10)	6 (4-9)	12 (7-23)	<0.001
Admission serum chloride (mmol/L)	104 ± 2	104 ± 2	103 ± 2	105 ± 2	104 ± 2	<0.001
Lowest serum chloride (mmol/L)	101 ± 3	102 ± 2	97 ± 3	103 ± 2	96 ± 3	<0.001
Highest serum chloride (mmol/L)	106 ± 4	105 ± 2	104 ± 2	111 ± 3	112 ± 3	<0.001

Continuous data are presented as the mean ± SD or median (interquartile range or IQR); categorical data are presented as count (%). * *p*-value for the comparison across four groups of in-hospital serum chloride levels was tested using an analysis of variance (ANOVA) for continuous data and the chi-square test for categorical data.

**Table 2 medicines-07-00038-t002:** The association between hospital-acquired dyschloremias and in-hospital mortality.

Serum Chloride During Hospitalization	N	In-Hospital Mortality	Univariable Analysis		Multivariable Analysis	
			OR (95% CI)	*p*	Adjusted OR (95 % CI)	*p*
Hospital-acquired hypochloremia						
No	28988	319 (1.1)	1 (ref)	-	1 (ref)	-
Yes	10310	169 (1.6)	1.50 (1.24–1.81)	< 0.001	0.88 (0.71–1.10)	0.27
Hospital-acquired hyperchloremia						
No	31458	190 (0.6)	1 (ref)	-	1 (ref)	-
Yes	7840	298 (3.8)	6.50 (5.41–7.81)	<0.001	2.50 (2.01–3.12)	<0.001
Groups						
Normal	23056	113 (0.5)	1 (ref)	-	1 (ref)	-
Only hypochloremia	8402	77 (0.9)	1.88 (1.40–2.51)	<0.001	0.91 (0.67–1.23)	0.54
Only hyperchloremia	5932	206 (3.5)	7.30 (5.80–9.20)	<0.001	2.84 (2.20–3.68)	<0.001
Both hypo- and hyperchloremia	1908	92 (4.8)	10.29 (7.78–13.60)	<0.001	1.72 (1.20–2.47)	0.004

Adjusted for age, sex, race, principal diagnosis, Charlson comorbidity score, coronary artery disease, congestive heart failure, peripheral vascular disease, stroke, diabetes mellitus, chronic obstructive pulmonary disease, cirrhosis, baseline glomerular filtration rate (eGFR), intensive care unit admission, number of hospital serum chloride measurements, length of hospital stay, and admission serum chloride levels. OR: odds ratio and CI: confidence interval.

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
