# Peer review of "Hospital-Acquired Serum Chloride Derangements and Associated In-Hospital Mortality"

_medicines, 2020, doi:10.3390/medicines7070038_

Round 1
Reviewer 1 Report
To the Editor,
I read the manuscript by Thongprayoon et al. titled:"Hospital-Acquired Serum Chloride Derangements and Associated In-Hospital Mortality". This is a well written paper which describes a retrospective analysis of patients admitted to a large tertiary center, and examines the association between hypo- and hyperchloremia and in-patient mortality. The manuscript main strengths are with the high number of patients, and the fact that the authors analyzed all-comers and not just critically ill patients. The main obvious limitation is the retrospective nature of the study. However, it does deliver important results that are needed for the understanding of dyschloremia and its relevance to patient outcomes.
There are several issues that the authors may change, which, in my mind, could improve the manuscript:
- Introduction: there is a lot of literature that suggests a causative link between hyperchloremia and AKI. It is mentioned often (including in papers cited in this work) that hyeprchloremia may cause poor outcome via AKI and its sequalle. I think this should be mentioned in the introduction in some way.
- Methods: the authors write the criteria for AKI and cite the KDIGO paper. Suggest to spell out in the text that they followed the KDIGO criteria and not just mention the creatinine threshold. On the same topic, it is not clear how the authors determined what is a patient's baseline creatinine. Is it the first value? what if the patient arrived in AKI?
- Methods: in the statistical analysis the authors state that they used AKI as a covariate for the regression analysis. Coming back to the first point, if AKI could be a result of hyperchloremia, why did they add it as a covariate?
- Results: If possible, it would be beneficial to spell out how many patients were screened, and how many and why were they excluded.
- Results (table 1): There is no mention as to what is the p-value referring to. Is it a comparison between two specific columns/groups? which ones? if it is between all four groups, ANOVA test with a post-hoc analysis is more appropriate than t-test.
- Results: could the authors mention what proportion of patients needed an admission to the ICU?
- Results (section 3.2): If I understand correctly, the authors did two types of comparisons: one in which hypo or hyperchloremic patients were compared to all the rest (so when comparing hypochloremic vs. non-hypochloremic, the later included the hyperchlroemic patients). Later they separated the patients into the four groups (persistently normal, just hypo, just hyper and both). In the way it is written, the difference between the two comparisons is not completely clear. Suggest to spell out that in the first type of comparison that in the non-hypo, for example, there are hyperchloremic patients.
- Results (section 3.2): in the same analysis, the authors report that hypochlroemia was not associated with mortality and report only the p-value (line 127). In the next paragraph, the report the OR and p-value (line 131). Suggest to report the OR for the non-statistically significant result as well.
- One of the things that is missing, in my opinion, is an analysis to check the temporal relation between dyschloremia and AKI. If the authors could find out whether dyschloremia occurred prior to AKI or not (and in what proportion), that will help shed more light on this complicated relationship. If a significant proportion of patients will demonstrate that dyschloremia happened before AKI, the authors should really consider removing AKI from their regression model. They could consider adding ICU admission as an alternative covariate.
- The discussion is well written yet may need to be adjusted if the authors would follow the suggestions above and results would change.
Author Response
Response to Reviewer#1
I read the manuscript by Thongprayoon et al. titled:"Hospital-Acquired Serum Chloride Derangements and Associated In-Hospital Mortality". This is a well written paper which describes a retrospective analysis of patients admitted to a large tertiary center, and examines the association between hypo- and hyperchloremia and in-patient mortality. The manuscript main strengths are with the high number of patients, and the fact that the authors analyzed all-comers and not just critically ill patients. The main obvious limitation is the retrospective nature of the study. However, it does deliver important results that are needed for the understanding of dyschloremia and its relevance to patient outcomes.
There are several issues that the authors may change, which, in my mind, could improve the manuscript:
Response: We thank you for reviewing our manuscript and for your critical evaluation.
Comment #1
Introduction: there is a lot of literature that suggests a causative link between hyperchloremia and AKI. It is mentioned often (including in papers cited in this work) that hyeprchloremia may cause poor outcome via AKI and its sequalle. I think this should be mentioned in the introduction in some way.
Response: We agree with the reviewer’s important comment. The following statements have been added to the introduction, as suggested.
The impact of hyperchloremia on renal function has been particularly considered for several years. The previous studies revealed that hyperchloremia resulted in renal vasoconstriction through tubuloglomerular feedback and increase of thromboxane release (7, 8). Recent data has demonstrated that dyschloremia is associated with worse patient outcomes, including acute kidney injury and longer hospital length of stay (3-6, 9-12).
Comment #2
Methods: the authors write the criteria for AKI and cite the KDIGO paper. Suggest to spell out in the text that they followed the KDIGO criteria and not just mention the creatinine threshold. On the same topic, it is not clear how the authors determined what is a patient's baseline creatinine. Is it the first value? what if the patient arrived in AKI?
Response: The reviewer raises important comment. We agree and the following statements have been revised to describe the definition of acute kidney injury and baseline serum creatinine.
Acute kidney injury (AKI) was defined according to KDIGO criteria as an increase in serum creatinine of ≥ 0.3 mg/dL or ≥1.5 times from baseline value (19), while the admission serum creatinine was used as the baseline serum creatinine
Comment #3
Methods: in the statistical analysis the authors state that they used AKI as a covariate for the regression analysis. Coming back to the first point, if AKI could be a result of hyperchloremia, why did they add it as a covariate?
Response: We agree with the reviewer. We removed acute kidney injury from the covariate adjustment as suggested.
Comment #4
Results: If possible, it would be beneficial to spell out how many patients were screened, and how many and why were they excluded.
Response: The following statements have been added to the result to describe the study selection process.
During the study period, 76,719 hospitalized patients with available serum chloride measurement at hospital admission were screened. After excluding patients with only one serum chloride measurement during hospitalization (n=18,839), and patients with abnormal admission serum chloride level (n=18,582), a total of 39,298 patients with normal serum chloride levels at hospital admission were studied.
Comment #5
Results (table 1): There is no mention as to what is the p-value referring to. Is it a comparison between two specific columns/groups? which ones? if it is between all four groups, ANOVA test with a post-hoc analysis is more appropriate than t-test.
Response: The reviewer raises important point. We added the following statements in the footnote of Table 1 to clarify p-value.
p-value for the comparison across four groups of in-hospital serum chloride levels was tested using analysis of variance (ANOVA) for continuous data, and Chi-squared test for categorical data.
Comment #6
Results: could the authors mention what proportion of patients needed an admission to the ICU?
Response: We appreciate the reviewer’s important comment. The proportion of patients needed ICU admission has been added to Table 1. The following statements have been added to the result section.
27% of patients needed ICU admissions during hospitalization.
Comment #7
Results (section 3.2): If I understand correctly, the authors did two types of comparisons: one in which hypo or hyperchloremic patients were compared to all the rest (so when comparing hypochloremic vs. non-hypochloremic, the later included the hyperchlroemic patients). Later they separated the patients into the four groups (persistently normal, just hypo, just hyper and both). In the way it is written, the difference between the two comparisons is not completely clear. Suggest to spell out that in the first type of comparison that in the non-hypo, for example, there are hyperchloremic patients.
Response: We revised these statements to clarify that patients with non-hypochloremia might include hyperchloremic patients, as well as patients with non-hyperchloremia might include hypochloremic patients, as suggested.
Comment #8
Results (section 3.2): in the same analysis, the authors report that hypochlroemia was not associated with mortality and report only the p-value (line 127). In the next paragraph, the report the OR and p-value (line 131). Suggest to report the OR for the non-statistically significant result as well.
Response: We agree with the reviewer. We added the odds ratio in the statement as suggested.
Comment #9
One of the things that is missing, in my opinion, is an analysis to check the temporal relation between dyschloremia and AKI. If the authors could find out whether dyschloremia occurred prior to AKI or not (and in what proportion), that will help shed more light on this complicated relationship. If a significant proportion of patients will demonstrate that dyschloremia happened before AKI, the authors should really consider removing AKI from their regression model. They could consider adding ICU admission as an alternative covariate.
Response: The reviewer raises important point. The relation in the timing of dyschloremia and acute kidney injury is very complex and we did not have this information. However, we agree with your comment that acute kidney might mediate higher dyschloremia-associated mortality. Therefore, we remove acute kidney injury from the multivariable regression model, but added ICU admission into the model instead, as suggested.
Comment #6
The discussion is well written yet may need to be adjusted if the authors would follow the suggestions above and results would change.
Response: After modifying multivariable model as suggested, the association of in-hospital dyschloremia and mortality remain consistent, suggesting the robustness of the result.
All authors thank the Editors and reviewers for their valuable suggestions. The manuscript has been improved considerably by the suggested revisions.

Reviewer 2 Report
Authors present quite interesting study about in hospital/total chloride concentration in the serum and its relation with overall mortality. Similar to other studies Authors show that intrahospital hyperchloremia can be related with higher patients mortality (Petnak T, Thongprayoon C, Cheungpasitporn W, Bathini T, Vallabhajosyula S, Chewcharat A, Kashani K. Serum Chloride Levels at Hospital Discharge and One-Year Mortality among Hospitalized Patients. Med Sci (Basel). 2020 May 19;8(2):E22. doi: 10.3390/medsci8020022; Ditch KL, Flahive JM, West AM, Osgood ML, Muehlschlegel S. Hyperchloremia, not Concomitant Hypernatremia, Independently Predicts Early Mortality in Critically Ill Moderate-Severe Traumatic Brain Injury Patients. Neurocrit Care. 2020 Feb 10. doi: 10.1007/s12028-020-00928-0).
Can Authors please explain why lower GFR level in patients with hyperchloremia was not related with AKI? (Table 1)? Adversely, more patients with hypochloremia had cardiovascular problems, but it did not seem to have influence on total mortality?
Author Response
Response to Reviewer #2
Authors present quite interesting study about in hospital/total chloride concentration in the serum and its relation with overall mortality. Similar to other studies Authors show that intrahospital hyperchloremia can be related with higher patients mortality (Petnak T, Thongprayoon C, Cheungpasitporn W, Bathini T, Vallabhajosyula S, Chewcharat A, Kashani K. Serum Chloride Levels at Hospital Discharge and One-Year Mortality among Hospitalized Patients. Med Sci (Basel). 2020 May 19;8(2):E22. doi: 10.3390/medsci8020022; Ditch KL, Flahive JM, West AM, Osgood ML, Muehlschlegel S. Hyperchloremia, not Concomitant Hypernatremia, Independently Predicts Early Mortality in Critically Ill Moderate-Severe Traumatic Brain Injury Patients. Neurocrit Care. 2020 Feb 10. doi: 10.1007/s12028-020-00928-0).
Response: We thank you for reviewing our manuscript and for your critical evaluation. We appreciate your important input. We have also included the suggested reference in our revised manuscript and found them very helpful.
Comment #1
Can Authors please explain why lower GFR level in patients with hyperchloremia was not related with AKI? (Table 1)?
Response: The reviewer raises important comment. Compared to patients with persistently normal serum chloride level, patients with hyperchloremia had higher acute kidney injury (18% vs. 7%; p<0.001). Hyperchloremia remain significantly associated with higher risk of acute kidney injury (OR 2.30; 95% CI 2.11-2.50, after adjusting for eGFR. In addition, lower eGFR was associated with higher risk of AKI (OR 1.16; 95% CI 1.15-1.17 per 10 ml/min/1.73 m2 increase).
Comment #2
Adversely, more patients with hypochloremia had cardiovascular problems, but it did not seem to have influence on total mortality?
Response: We appreciate the reviewer’s important comment. In unadjusted analysis, hypochloremia was associated with higher mortality. However, adjusting for confounders, including cardiovascular problems, the association of hypochloremia and mortality was not significant.
All authors thank the Editors and reviewers for their valuable suggestions. The manuscript has been improved considerably by the suggested revisions.
